# Mapping and Analysing Potential Sources and Transmission Routes of Antimicrobial Resistant Organisms in the Environment using Geographic Information Systems—An Exploratory Study

**DOI:** 10.3390/antibiotics8010016

**Published:** 2019-02-27

**Authors:** Carlos Chique, John Cullinan, Brigid Hooban, Dearbhaile Morris

**Affiliations:** 1Discipline of Economics and Health Economics and Policy Analysis Centre, National University of Ireland, Galway, H91 CF50, Ireland; carlos.chique@nuigalway.ie; 2Discipline of Bacteriology, School of Medicine and Centre for Health from Environment, Ryan Institute, National University of Ireland, Galway, H91 CF50, Ireland; b.hooban1@nuigalway.ie (B.H.); dearbhaile.morris@nuigalway.ie (D.M.)

**Keywords:** antimicrobial resistance, antimicrobial resistant organisms, sources, transmission routes, GIS, Ireland

## Abstract

Antimicrobial resistance (AMR) is one of the leading threats to human health worldwide. The identification of potential sources of antimicrobial resistant organisms (AROs) and their transmission routes in the environment is important for improving our understanding of AMR and to inform and improve policy and monitoring systems, as well as the identification of suitable sampling locations and potential intervention points. This exploratory study uses geographic information systems (GIS) to analyse the spatial distribution of likely ARO sources and transmission routes in four local authority areas (LAAs) in Ireland. A review of relevant spatial data in each LAA, grouped into themes, and categorised into sources and transmission routes, was undertaken. A range of GIS techniques was used to extract, organise, and collate the spatial data into final products in the form of thematic maps for visual and spatial analysis. The results highlight the location of ‘clusters’ at increased risk of harbouring AMR in each LAA. They also demonstrate the relevance of aquatic transmission routes for ARO mobility and risk of human exposure. The integration of a GIS approach with expert knowledge of AMR is shown to be a useful tool to gain insights into the spatial dimension of AMR and to guide sampling campaigns and intervention points.

## 1. Introduction

The World Health Organisation (WHO) has highlighted antimicrobial resistance (AMR) as one of the most substantial threats to human health worldwide, with a growing number of critical infections becoming increasingly difficult to treat with the current line of antibiotics [1]. It is estimated that by 2050, unless action is taken, 10 million deaths per year will be attributable to AMR [2]. Although a substantial amount of research has focused on the rapid transmission of antimicrobial resistant organisms (AROs) within the clinical environment, the natural environment is also an important reservoir, not only for facilitating the spread of AROs, but also as a point of contact for humans and animals to become colonised or infected [3].

The production of antibiotics is a natural occurrence by environmental organisms, hypothesised to be a means of communication [4], and many mechanisms of resistance are embedded in basic survival components of the bacteria. These include efflux pumps, which function in the detoxification of bacterial cells from products of their own metabolism, as well as antibiotics [5]. Although these natural mechanisms contribute to a certain level of resistance in the environment, the anthropogenic introduction of semi-synthetic antibiotics, especially in hotspot areas such as hospitals or large agricultural regions, amplifies this phenomenon [6]. The concern associated with the introduction of antibiotics into the environment stems from the fact that although low concentrations of antibiotics may not be capable of killing the bacteria, they do employ a selective pressure that encourages them to adapt [7]. This confers a survival advantage over susceptible isolates and enables them to multiply in numbers, facilitating their transmission and further dissemination of resistance genes. Over time, resistance genes become widespread among bacteria through repeated exposure to sub-therapeutic levels of antibiotics.

Bacteria may be intrinsically resistant to antimicrobial agents or may acquire resistance as a consequence of mutation or horizontal acquisition of antimicrobial resistance genes (ARGs) frequently located on mobile genetic elements, for example, plasmids, allowing for rapid transfer of resistance determinants between bacteria of different species and within different environmental niches. There are a number of different types of AROs, some of which are resistant to the last resort antibiotics, for example, the carbapenemase producing *Enterobacteriaceae* (CPE). In many cases of infection with CPE, there are only three antimicrobial agents available for treatment, namely, colisitin, tigecycline, and fosfomycin [8]. Unfortunately, reports of resistance to these last resort agents are emerging, including the recently reported findings of plasmid-encoded colisitin resistance (*mcr-1*), initially in pigs in China, and subsequently in humans and animals worldwide [9]. If this situation continues, we may enter an era of untreatable infections, and it is thus important that a ‘One-Health’ approach is adopted to tackle the problem of AMR, that is, an approach that recognises that the health of humans, animals, and the environment are interconnected.

Thus, the emergence and dissemination of AMR is related to use of antimicrobial agents, which have been used for decades in human and animal medicine and for other applications. It is only recently that attention has been given to the potential impact the release of such products after use has on the natural environment and the pivotal role the environment plays in the persistence and spread of AMR. The major sources of antimicrobials, AROs, and ARGs in the environment include human and animal waste, inappropriate disposal of unused antimicrobial agents, and effluent from facilities manufacturing antimicrobial agents. Furthermore, the aquatic environment is an important potential transmission route of AMR to humans, animals, and the food chain. Surface waters are often discharge points for wastewater or runoff from agricultural land, while also serving as sources of drinking water supplies and/or waters used for recreational purposes.

Within this context, this paper is concerned with the identification of sources of AROs and their transmission routes in the environment, in order to improve our understanding of AMR and to inform and improve policy and monitoring systems. In particular, we look to map and analyse the sources and transmission routes of AROs in the environment using geographic information systems (GIS). This is done for four local authority areas (LAAs) in Ireland; namely, Cork County Council, Fingal County Council, Galway City Council, and Galway County Council. These LAAs were selected on the basis of geographical spread, the likely presence and diversity of potential sources of AROs in the environment, as well as being representative of the national picture. The LAAs selected also represent adequately-sized and manageable units to perform GIS analysis.

The overall goal of this study is to explore the spatial and environmental dimensions of AROs. The implementation of geospatial analysis in health research is important for recognising and highlighting the connection between human populations and relevant spatial factors at a ‘continuous’ landscape setting [10]. Through the establishment and visualization of spatial associations among incidence data, human population, and disease sources, GIS has the potential to increase our understanding of the prevalence, transmission, and ultimately prevention of several diseases [11].

The potential of a GIS approach in AMR research, grounded on landscape ecology principles and integrating key biotic and abiotic spatial variables, has been recognised for some time [4]. Even so, the application of geospatial techniques in AMR research remains relatively uncommon, though there have been studies that have focused on the spatial distribution of selected incidence data [12]. A holistic approach that incorporates potential ARO sources and transmission routes is important for improving our understanding of AMR–environment interactions [4], but, to our knowledge, there has been no comprehensive attempt to identify, map, and quantify ARO sources and/or transmission routes within a landscape context. Furthermore, GIS is rarely used as an initial screening tool to guide, for example, sampling campaigns, which can often be expensive and time-consuming. A significant advantage of GIS analysis is that it can assist researchers in integrating sources and transmission routes of AROs in the environment. Identified areas can then be targeted for comprehensive sampling.

Sources of AROs considered in this paper include ‘clusters’ of antibiotic usage in both urban and agricultural settings, while transmission routes refer to the ‘medium’ facilitating ARO mobility and expansion in the environment. A wide range of possible sources of AROs are mapped, including hospitals, long term care facilities (LTCFs), wastewater treatment plants (WWTPs), and so on. Data on antimicrobial use in hospitals and LTCFs are also incorporated, as is the census of population data on types of water supplies; census of agriculture data on farm intensity and livestock numbers; and data on raw sewage, septic tanks, landfill leachate, and so on. Potential routes of transmission to and within the environment and the potential for human exposure are also mapped, including surface water networks (rivers, lakes, estuaries, designated bathing areas), groundwater network and vulnerability, water supplies (public, group water schemes, private wells), as well as land use data. The justification for the specific sources and transmission routes included is set out in Section 2.

The maps developed in this paper are currently being used to inform sampling locations in an Environmental Protection Agency (Ireland) (EPA) funded project titled ‘AREST’, which looks to identify areas harbouring fluoroquinolone-resistant extended spectrum beta lactamase (ESBL) and CPE. This paper draws on work from the project and, in this regard, the utility of a GIS approach in overlaying layers of spatial data is demonstrated by integrating a range of different sources with a variety of transmission routes. This is relevant because ARO mobility in aquatic environments is likely to be important for the spread of AMR. The benefits of using GIS as a database to store, analyse, share, and visualise spatial data are also demonstrated. More specifically, we illustrate how GIS interface data can be displayed interchangeably with ease to evaluate different criteria, scenarios, and pressures. Overall, the results of this investigation are expected to improve our understanding of AMR–environment dynamics to inform and improve policy and monitoring systems.

## 2. Identifying Sources and Transmission Routes

The first step in this exercise was to identify the relevant variables that should and could be mapped. To this end, we began by undertaking a comprehensive search for all relevant spatial data in each of the four LAAs, grouped into themes, and categorised into sources and transmission routes. This search was aided by a thorough review of the literature on the sources and transmission routes of AROs, which is now summarised.

### 2.1. Sources

There are many anthropogenic sources that may contribute to the accumulation of different forms of AMR in the environment. The results of a recent systematic literature review [13] indicate that WWTPs are the most studied point source of AROs, with high numbers of ARGs observed downstream of rivers receiving WWTP effluent. This contrasts to upstream river sections or sites in the near vicinity of the plants, which feature lower numbers of ARGs. The implication is that areas surrounding WWTP discharge points are hotspots for resistant organisms.

Industrial emissions from pharmaceutical production plants also feature as an important contributor to environmental AMR. A number of studies have demonstrated that samples taken downstream from WWTPs largely fed by effluent from pharmaceutical production plants have a significantly higher abundance of ARGs and higher resistance ratios in comparison with upstream samples [14,15,16]. These studies demonstrate the significant role the pharmaceutical industry plays in the environmental microbiome by releasing concentrated quantities of antibiotics.

Hospitals and LTFCs are known to have high antibiotic consumption rates and, as a result, those that feed into WWTPs significantly increase environmental resistance in the surrounding waters. Ludden et al. [17] tested treated and untreated wastewaters for CPE and observed that WWTPs located in close proximity to, and receiving wastewater from, hospitals typically tested positive for CPE. Data from the latest Healthcare Associated Infections and Antimicrobial Use in LTCFs (HALT) point prevalence survey in Ireland found that 1 in 10 residents of LTCFs were under antibiotic treatment, and this proportion increased to 1 in 3 residents in palliative care LTCFs [18]. A significant quantity of the antibiotics given to patients in healthcare institutions is shed into wastewater via urine or faeces, in a form that is still biologically active. Furthermore, a high proportion of patients have AROs resident in their gut, significant numbers of which ultimately enter the urban wastewater stream. According to recent research, current wastewater treatment processes do not successfully remove all AROs [19].

The use of antibiotics in veterinary medicine is also a significant contributor to environmental resistance and animal waste, both agricultural and wild, represents a risk for transmission and persistence of AMR in soils and the aquatic environment. Animal manure is recognised as an important reservoir of AROs and ARGs, both acquired and intrinsic resistances [20,21]. With increased productivity in the agricultural sector and planned increases in the national herds under Food Wise 2025 [22], there will be a significant increase in waste produced by the Irish agricultural sector (and elsewhere). The WHO recommends antibiotics should not be used as growth promoters or in cases where no diagnosis of bacterial infection has occurred in a member of the herd [23].

In Ireland, land spreading of organic waste including animal manures and sewage sludge is a common agricultural practice and an integral part of the ‘circular’ economy. There is increasing concern that such practices may contribute to the presence of ARGs in resident soil bacteria and AROs in soil [24]. AROs and ARGs in soil can enter the food chain via contaminated crops or groundwater, and may impact human health. A recent study reported that land spreading involving dairy cattle manure enhanced the proliferation of resident AROs and ARGs in soils, even though cattle from which the manure was collected were not being treated with antimicrobial agents [24].

### 2.2. Transmission Routes

The aquatic environment represents a crucial and often overlooked potential transmission route of AMR to humans and animals. Surface waters often serve as discharge points for both wastewater and runoff from agricultural land and urban areas, while also serving as sources of drinking water supplies and/or water used for recreational purposes. A recent systematic literature review by Leonard et al. [25] demonstrated an increased risk of acquiring an infection in relation to seawater exposure. The studies included in the review compared non-bathers with bathers to demonstrate the statistically significant difference in terms of infections between the two groups. Similarly, another review by Leonard et al. [3] also highlighted the level of risk of exposure to antibiotic resistant bacteria in coastal waters and its relationship to different types of water sports. In Ireland, there are currently 38 locations where wastewater is being discharged into surface waters without any form of treatment [26]. This represents a very significant risk for transmission and persistence of AMR in the environment, as evidenced by recent findings of CPE in bathing waters attributable to raw sewage discharging to these waters [27].

Apart from recreational use, the aquatic environment plays a vital role in supplying water for human consumption. Water for domestic consumption is commonly extracted from groundwater via wells and boreholes. These sources are at potential risk of microbial contamination if in close proximity to septic tanks or as a result of agricultural activities such as land spreading of animal manure [28,29]. Groundwater contamination is a particularly recurrent feature in an Irish context, with recent data from the EPA indicating that 42% of groundwater monitoring sites (195) tested positive for the presence of *E. coli* [30]. Therefore, overall, it is important that the natural aquatic environment is considered for its role in the dissemination of AMR and that efforts are put in place to minimise contamination.

## 3. Materials and Methods

### 3.1. Materials

#### 3.1.1. Data Review and Selection

Following consideration of the relevant literature as outlined above, a comprehensive review of relevant spatial data in Ireland, grouped into themes (e.g., healthcare, agriculture), and categorised into ARO sources and transmission routes, was undertaken (see Table 1). The data selection process was also supported by expert knowledge in the form of the AREST Project Team. Spatial datasets were obtained from several Irish authorities (e.g., Environmental Protection Agency (EPA), Central Statistics Office (CSO)) or independently produced through manual geolocation and geocoding. Datasets include GIS-compatible files (Shapefiles) of feature classes—or spatial data layers—in vector format (point, multipoint, line, and polygon features), as well as attribute data (i.e., non-spatial information relating to spatial features), which was modified, related, and incorporated into feature classes in a GIS environment. All datasets selected represent the most up-to-date versions available.

The bulk of the data, in terms of spatial data layers representing ARO sources and transmission routes, was obtained from the EPA. In turn, the majority of attribute data (e.g., demographics, agriculture) was retrieved from CSO census estimates [31,32]. The EPA also provided access to a range of datasets pertaining to the environmental quality/status and key human pressures exerted on transmission routes. These include inland waters (rivers/lakes), marine environments (coastal/transitional waters), and groundwater bodies (Table 1). Several of these datasets, and associated estimates and classifications, were established in the context of legislation at European and Irish levels, including the Water Framework Directive (WFD) [33] and the Water Services (Amendment) Act (S.I. No. 2 of 2012). The EPA enforces these statutes through the WFD Monitoring Programme and the National Inspection Plan (NIP) [34,35]. Their incorporation in the mapping exercise was based on their potential to inform and guide the sampling site selection process in the AREST project.

#### 3.1.2. Antimicrobial Usage Data

Rates of antimicrobial use in Irish healthcare facilities were retrieved from the results of point prevalence surveys (PPS). The European healthcare-associated infections and antibiotic use in long-term care facilities (HALT) survey [18] provided antimicrobial use rates for LTCFs. Rates were calculated as antimicrobial applications per occupied bed units (AMA per OBU) on the day of the survey. Antimicrobial use rates in public general/acute hospitals were obtained from the HPSC PPS for the year 2017 [36]. Total hospital antimicrobial consumption is defined as daily doses per 100 bed days used (DD per 100 BDU). Antimicrobial usage rates (as attribute data) were related to corresponding spatial features (healthcare facilities) in each LAA.

### 3.2. Methods

#### 3.2.1. Spatial Mapping, Data Analysis, and Geolocation/Geocoding

All spatial and data analyses were performed within a GIS environment (ESRI^®^ ArcGIS^®^ ArcMap™ version 10.2). The ungeneralised (high-resolution) administrative boundary Shapefile for Ireland from the CSO [37] was used to delineate relevant LAAs and create independent ‘base maps’ for each area. Each LAA base map is composed of electoral divisions (EDs) (polygons), which comprise the smallest legal administrative division in Ireland (*n* = 398, County Cork; *n* = 236, County Galway; *n* = 22, Galway City; *n* = 42, Fingal County). A range of key public information in Ireland, including demographics and agricultural census data, is available at the ED level. Generally, ED unit size is proportionate to local population numbers and they are generally moderate–small in size. As such, EDs form the spatial basis facilitating ARO source and transmission route integration and map visualization. This approach was fundamental for the identification and categorization of sampling areas with manageable dimensions. Independent base maps were used to correlate feature classes to their corresponding geographic location in each LAA and ED.

All spatial data layers obtained were standardised into a single projected coordinate system (IRENET95 Irish Transverse Mercator) using the Feature Project tool in ArcMap. A range of ArcMap tools and functions (e.g., Clip, Union, Merge, Table Join, Dissolve) were used to modify, organise, and collate all datasets. The Spatial Join and Calculate Geometry functions were used to quantify potential ARO sources at the ED level, as well as to estimate the area of groundwater bodies and number of boreholes under specific categories following EPA National Inspection Plan (NIP)-Water Framework Directive (WFD) classifications. Certain feature classes were supplemented or entirely created through manual geolocation and geocoding. This initially involved obtaining corresponding geographic coordinates based on postal address data through online batch geocoding. Features were then manually identified on imagery within the Google Earth Pro (ver 7.3.2) platform and batch exported (as KML files) into ArcMap. The KML to Layer conversion tool was then used to create GIS Shapefiles of feature classes.

#### 3.2.2. Data Standardization

A process of data standardization was required to simplify and harmonise feature classes across all LAAs. While standardization involved several datasets, the process particularly pertained to those comprising commercial and industrial activities registered with the EPA in each LAA (e.g., Section 4 Discharges, Industrial Emissions). The process entailed classifying site activity records into pre-defined categories. Section 4 Discharges refers to EPA-registered commercial activities discharging effluents into surface water. In this instance, facilities/effluents were grouped into healthcare, animal/food production, and food/hospitality categories based on individual EPA application records. The standardization task also involved exclusion of data within feature classes to simplify datasets and analyses. For example, the Geological Survey Ireland (GSI) groundwater well dataset was modified to only include boreholes used in the extraction of groundwater for domestic and/or agricultural consumption, that is, potential exposure of AROs in domestic and agricultural settings.

#### 3.2.3. Estimating Livestock ‘Intensity’—Generating a Livestock Weighted Index

As a key source of AROs, attempting to quantify levels of livestock presence in each LAA and ED was essential. The CSO agricultural census [31] provided cattle and sheep numbers at the ED level. Because of a confidentiality clause (small number of farm holdings per ED), numbers for pig and poultry were unavailable in all four LAAs. However, ED incidence data (presence/absence) for both livestock categories were made available by the CSO and incorporated into a feature class as attribute data. All livestock data were integrated into a single ‘index’ using the Weighted Attribute Overlay tool in ArcMap. The GIS tool allows users to assign numerical weights to individual categories in order produce a final attribute value based on selected inputs. Because of the lack of estimates on antibiotic applications in animals in Ireland, available data of antibiotic usage and sales in both European [38] and U.K. livestock [39], supported by local expert knowledge (AREST project team), provided feedback to discriminate among perceived levels of antimicrobial use in livestock categories. Hence, EDs with pig and poultry (high antimicrobial use) presence were assigned a higher weight, followed by cattle density and, lastly, sheep density with an almost neutral weight (low antimicrobial use). The final ED-level livestock index feature class was used as a base map layer overlaid by both ARO sources and transmission route data layers in several thematic cartographic displays.

#### 3.2.4. Cartographic Displays, Spatial Resolutions, and GIS Data Classification

Spatial overlaying, or superimposition of data layers, was used to create a range of thematic maps based on specific sampling criteria and targets. Cartographic displays integrate distinct ARO sources and transmission routes in order to identify different scenarios and human pressures at the landscape level. Through an iterative process, the GIS interface allows additional data layers to be incorporated onto existing maps to evaluate the relevance of different data parameters. This approach followed the combination of ARO sources with the three main categories of transmission routes: (i) hydrology (inland waters), (ii) marine environments (coastal/transitional waters), and (iii) groundwater bodies.

The identification of potential sampling areas is aided by visual inspection of maps at different spatial resolutions. A GIS interface allows users to apply interchangeable spatial scales of visualised data. Maps at the LAA (macro-) level were found to be most suitable in identifying main clusters of ARO sources. However, as a result of data congestion, adopting a fine-scale mapping resolution allowed for a more efficient integration of ARO sources and transmission routes. This is particularly relevant in the context of LAAs with larger dimensions (County Cork and County Galway). Employing interchangeable spatial scales also enabled different levels of data ‘classification’ in feature classes. Data layer classification refers to the level of visual information (derived from attribute data) displayed for an individual feature class. Accordingly, data layers in maps generated at the LAA level were simplified to circumvent data overcrowding, while a higher level of feature class detail was attainable in fine-scale maps. Using a finer spatial scale also provided the opportunity to include aerial imagery and road map elements (as base maps) onto cartographic displays to better interpret landscape settings and facilitate sampling efforts.

## 4. Results

### 4.1. ARO Sources, Livestock Weighted Index, and LAA Cartographic Displays

Summary statistics for potential sources of AROs, livestock index estimates, and healthcare facilities identified in the mapping exercise are presented in Table 2 for LAAs (and selected EDs). Overall, County Cork is the LAA with the highest number of potential ARO sources, a factor likely linked to a higher population and larger size area in comparison with the other LAAs. Similarly, County Cork is the LAA with the highest proportion of the livestock index estimates in the ‘high’ and ‘medium’ classifications, at 16% and 47% of EDs, respectively. This is followed by County Galway (high = 10%; medium = 40%) and Fingal County (high = 4%; medium = 16%)—see Table 2.

Figure 1 provides a simplified illustration of identified ARO sources and livestock estimates in each LAA. This unclassified (other than healthcare facilities) map visualization provides a preliminary and generalised overview of ARO sources useful in the identification of spatial clustering patterns. Clusters tend to show a clear association with urbanised areas in all maps. For example, Figure 1A shows that Cork City (which is not part of the AREST study area) and its environs (which are part of the AREST study area) display a large concentration of potential ARO sources, while Figure 1B shows a similar situation for Galway City and Ballinasloe, including the presence of county hospitals (Figure 1B). ARO sources show a relatively more even distribution in Fingal County, but a higher concentration is recorded at Blanchardstown and The Ward (south-west), including several LTCFs and one hospital, in close proximity to Dublin City Centre (south of Fingal) (Figure 1C). Livestock estimates tend to be higher in rural areas that often exhibit low numbers of other ARO sources. This has implications in terms of sampling criteria, area prioritization, and potential trade-offs between the two (see ED values in Table 2).

While unclassified maps are useful in identifying clusters and potential sampling areas, the cartographic displays presented in Figure 2, Figure 3 and Figure 4 provide a higher level of detail through source classification. Classified maps enable us to discriminate among clusters based on sampling criteria and allow for informed decision making. For instance, the bulk of ARO sources at Lehenagh, the ED with the largest concentration among the LAAs (*n* = 33) (Table 2), are waste emission points from a landfill site (Figure 2). Sampling within the ED would be practical if following specific sampling criteria, but the lack of diversity in ARO sources also makes the area unattractive for more comprehensive sampling efforts. Similarly, a significant proportion of ARO sources in Blanchardstown (Fingal County) comprise pharmaceutical industrial emissions (Figure 4), which could be a setting targeted to investigate the contribution of the pharmaceutical industry towards ARO environmental prevalence. Overall, cartographic displays at the LAA level are valuable, but are still constrained by excessive data aggregation and impaired visualization. Inclusion of transmission routes at the LAA level, which is key to assess and understand ARO mobility at the landscape level, exacerbates this data congestion.

### 4.2. Fine-Scale Cartographic Displays

Figure 5, Figure 6, and Figure 7 provide examples of fine-scale displays in Cork Harbour (County Cork), a highly relevant area considering the number and diversity of ARO sources. Fine-scale maps enable a higher level of data layer classification with the GIS interface, allowing for the inclusion/exclusion of feature classes based on relevant sampling criteria with ease. Transmission routes are efficiently incorporated into cartographic displays at this scale, facilitating considerations of sources of AROs and potential mobility within aquatic environments. In Figure 5, Cork Harbour is shown to be a potentially interesting sampling area with several raw sewage discharges and a high number of urban wastewater points discharging into coastal waters. Several local EDs have ‘medium’ livestock estimates with coastal/transitional water bodies under significant stress by cultural activity (urban waste water, agriculture and anthropogenic pressures) following EPA-WFD guidelines (Figure 6). Figure 7A,B provide a ‘zoom-in’ example on a potential sampling area in coastal waters near Haulbowline Island in close proximity to a large raw sewage discharge, a landfill site, and several waste/urban emissions. Inclusion of satellite imagery at this level aids in landscape interpretation, allowing the identification of relevant features (e.g., urban fabric, farmland, river dimensions) and, in combination with road map elements, facilitates planning for sampling campaigns, such as identifying road access to sampling sites. Fine-scale cartographic displays using a range of data layer combinations were generated for all main clusters of ARO sources identified (Figure 1, Figure 2 and Figure 3).

### 4.3. Examples of Data Layer Combinations

Table 3 summarises some of the key data layer combinations and thematic maps generated through the mapping exercise, while Figure 8, Figure 9 and Figure 10 provide examples of map interpretation and potential applications at the LAA level. As highlighted above, maps at the LAA level are most useful to identify areas of interest, which can then be evaluated in more detail through fine-scale cartographic displays.

Figure 8 incorporates all ARO sources and surface water transmission routes (inland waters, marine environments) identified in Fingal County. In this example, consideration of the bathing site location in light of the distribution of ARO sources and coastal/transitional water status is of particular relevance because of the potential of recreational waters to act as hotspots of AMR transmission in humans [3,25]. The Rogerstown Estuary is shown as an area of potential interest, classified as a transitional water body under ‘agricultural’ pressures with river effluents draining EDs with ‘medium’ and ‘high’ livestock estimates. Inclusion of the EPA-WFD significant pressures on rivers layer serves to verify that effluents are indeed subject to pollution from agricultural emissions. Additionally, two landfill facilities and one waste emission point to surface water are located within the estuary. While currently classified as having ‘excellent’ water quality, bathing sites at Loughshinny, Portrane, and Rush are under restriction based on poor water quality performance in the year 2017. These locations are also situated in close proximity to urban wastewater discharges and coastal water under EPA-WFD ‘anthropogenic’ pressures. Following EPA criteria, ‘anthropogenic’ pressures include microbiological, organic, sediment, and nutrient pollution. As such, these sites could be considered suitable targets to investigate ARO incidence in both coastal and transitional waters.

Contaminated drinking water also represents an important source of AROs and human transmission [40]. A number of thematic maps analysing the distribution of DWTPs and pollution sources were created. Figure 9 illustrates one such example incorporating the location of DWTPs, EPA-WFD significant surface water pressures (lakes and rivers), and selected wastewater ARO sources in County Cork. Visual inspection indicates that DWTPs serving Mallow and Macroom are situated in areas with a high prevalence of urban wastewater discharges. The Mitchelstown DWTP is located in a landscape with high livestock density with a significant pig farming industry—see also Figure 2. Additionally, the DWTP at Inniscarra is fed by lake water under anthropogenic pressures. The information derived from these maps can be used in conjunction with DWTP performance reports and plant water treatment specifications to identify suitable sampling sites.

In contrast to surface water, access to groundwater wells (boreholes) and/or monitoring stations is required to test the prevalence of AROs in groundwater. Accordingly, several thematic maps incorporating the location of boreholes, EPA groundwater monitoring stations, and variables associated with groundwater status (ARO sources, EPA NIP-WFD classifications) as target criteria were generated. Visual (map) interpretations were supplemented by calculations of groundwater area extension under specific EPA classifications and the number of boreholes situated within each category (Table 4). Overall, the results reflect the intrinsic geological properties of County Galway and County Cork with soils highly susceptible to groundwater contamination by human activity [41]. Groundwater sampling is also readily accessible in these two LAAs with estimates of boreholes per category expediting site selection among LAAs. Figure 10 aids in visualising some of this information including rates of groundwater vulnerability to human contamination, (ED) septic tank prevalence, boreholes, and selected wastewater ARO sources in County Cork. The purpose of the thematic map is to identify boreholes with an increased likelihood of harbouring AROs based on rates of soil permeability, ARO sources, and potential of septic tank seepage. Similar to other LAAs, boreholes tend to be concentrated outside urban centres, where ARO sources tend to be lower. Even so, a number of ‘focus’ areas characterised by high groundwater vulnerability (i.e., bare rock/karst and extreme/high categories), high incidence of septic tanks (>50% households per ED), and aggregation of boreholes (including diverse source use types) were identified. These areas can then be analysed in more detail through fine-scale mapping. Substituting the groundwater vulnerability base map with other EPA NIP-WFD groundwater base map layers (see Table 4), different sets of ARO sources, and/or different livestock estimates allows for the consideration of additional criteria and scenarios in a practical and efficient manner.

## 5. Discussion and Conclusions

AMR is a leading threat to human health and the identification of sources of AROs and their transmission routes in the natural environment is important for improving our understanding of AMR, for informing and improving policy and monitoring systems, and for the identification of suitable sampling locations and potential intervention points. This paper presents an exploratory study that uses GIS to analyse the spatial distribution of ARO sources and transmission routes in four LAAs in Ireland, and highlights the location of clusters at increased risk for persistence and transmission of AROs in each area.

The examples provided in this paper highlight the potential role for GIS as a tool aiding the identification of geographic areas that are populated with multiple, and often spatially complex, sources of AROs. GIS mapping allows for the combination of spatial data based on an array of criteria to identify potential clusters of sources and to assess how these might impact local populations. Beyond the identification and classification of sources, a GIS approach also facilitates the combination of variables that are typically related in the dissemination of AROs in the environment. For example, there are multiple variables that are known to affect the likelihood of groundwater contamination, including the proximity of septic tanks, animals or manure spreading, soil type, and groundwater vulnerability. GIS allows spatial information on these variables to be combined and visually inspected, in order to undertake an informed assessment of how likely a specific area is to be at risk of contamination. This ability to overlay (or discriminate) information is also advantageous when selecting sampling locations to screen for antibiotic resistance, which is a complicated decision requiring consideration of multiple attributable factors. In this regard, GIS enables a systematic approach to be applied to sampling in terms of deciding upon the most relevant areas to target. However, it is important to note that our approach has not yet been verified, as to date no sampling has been undertaken at any of the identified sources or transmission routes. This sampling is currently the focus of ongoing research.

Monitoring of the environment for AMR is an important area of research, as highlighted in Objective 2 of the WHO’s ‘Global Action Plan on Antimicrobial Resistance’ [1]. WHO guidelines outline the need for a greater understanding of how resistance spreads by adopting a One-Health approach to include humans, animals, and the environment. Therefore, it is important that environmental samples are screened for significant types of resistances that are currently prevalent and impacting the treatment outcomes of patients, such as CPE. Because the distribution of humans, animals, and environmental features is likely to be spatially uneven, adopting a GIS approach is particularly useful within a One-Health perspective. It is worth noting that GIS can also be used in other ways to help develop our understanding of AMR. For example, previous research has used GIS to explore the occurrence of AMR *E. coli* causing urinary tract infections in the community [42].

In terms of some of the practicalities involved in the mapping exercise illustrated in this paper, our approach highlights the importance of good availability of, and access to, spatial and other attribute data from various public authorities and institutions (e.g., EPA, CSO). In this regard, public agencies play a particularly important role in facilitating adequate data access for this type of analysis. This issue also highlights the importance of public data sharing schemes and initiatives. Nonetheless, limitations will likely exist both in terms of the range of variables that are available at a spatial level, as well as the detail associated with these variables. For example, in this mapping exercise, it was necessary to make some working assumptions in order to generate our livestock index because of data confidentiality measures in place. For other variables, geocoding was necessary to pinpoint the locations of certain sources. In some cases, it was not possible to ascertain the ‘ground’ accuracy of spatial datasets that were mapped. Accordingly, such features would need to be subject to field corroboration during sampling campaigns. Moreover, the required data are not always up-to-date. For example, our livestock index was based on census of agriculture data from 2010 as survey results are only publicly available in 10-year intervals.

Overall, such factors meant that collection, collation, processing, and mapping of the data were at times relatively time-consuming and labour intensive. However, the established GIS framework can now be easily updated once new spatial information becomes available, essentially acting as a digital database for the selected LAAs. The approach implemented also has a high potential for expansion at a national scale and can be easily adopted by relevant authorities to monitor and characterise AMR prevalence at manageable geographic units, for example, county level. This would ideally incorporate data on ARO prevalence and AMR incidence cases. Given that this was the first time such an exercise was attempted, this investigation also presents a useful template for conducting similar exercises in other countries with accessible spatial data in the future.

## Figures and Tables

**Figure 1 antibiotics-08-00016-f001:**
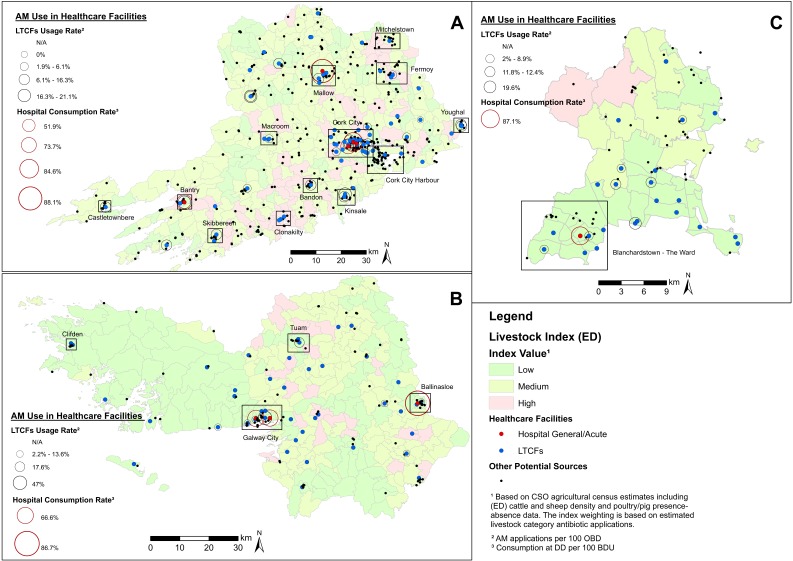
Composite map of potential antimicrobial resistant organisms (ARO) sources (unclassified) and healthcare facilities (including antimicrobial (AM) use) in County Cork, County Galway, and Fingal County. Notes: Livestock index estimates are included as a base map layer. Clusters of ARO sources in urban areas are delineated. LTCFs = long-term care facilities. ED = electoral district. CSO = Central Statistics Office.

**Figure 2 antibiotics-08-00016-f002:**
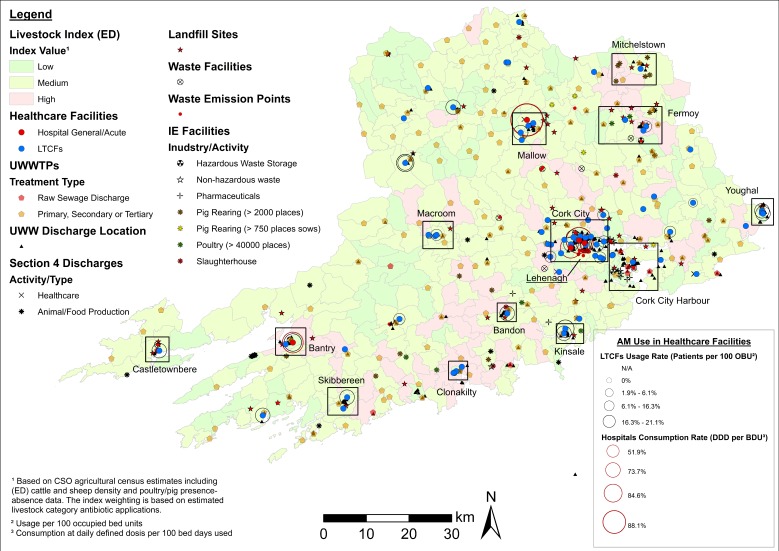
Map of potential ARO sources (classified) and healthcare facilities (including AM use) in County Cork. Notes: Livestock index estimates are included as a base map layer. Clusters of ARO sources are delineated with the location of Lehenagh—the ED with the highest number of ARO sources identified (highlighted). ED = electoral district. UWWTPs = urban wastewater treatment plants. UWW = urban wastewater. IE = industrial emissions. OBU = occupied bed units.

**Figure 3 antibiotics-08-00016-f003:**
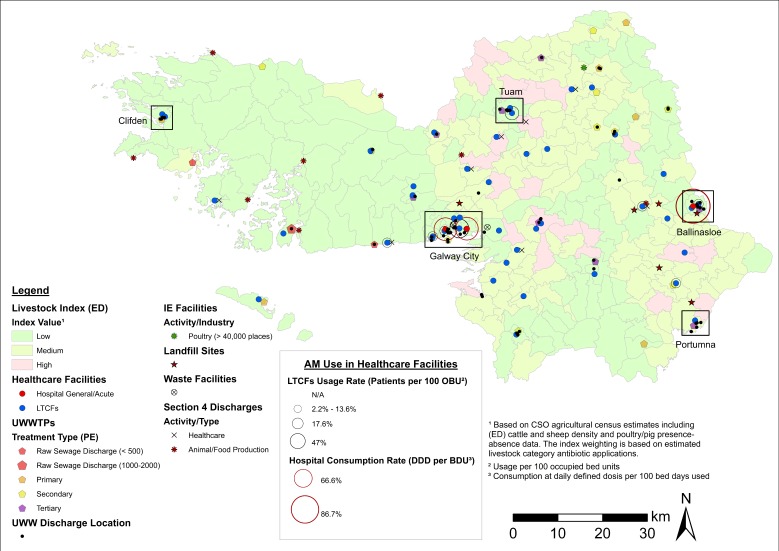
Map of potential ARO sources (classified) and healthcare facilities (including AM use) in County Galway (including Galway City). Notes: Livestock index estimates are included as a base map layer. Clusters of AROs sources in urban areas are delineated. ED = electoral district. UWWTPs = urban wastewater treatment plants. UWW = urban wastewater. IE = industrial emissions. PE = population equivalent (i.e., estimated population served by UWWTP).

**Figure 4 antibiotics-08-00016-f004:**
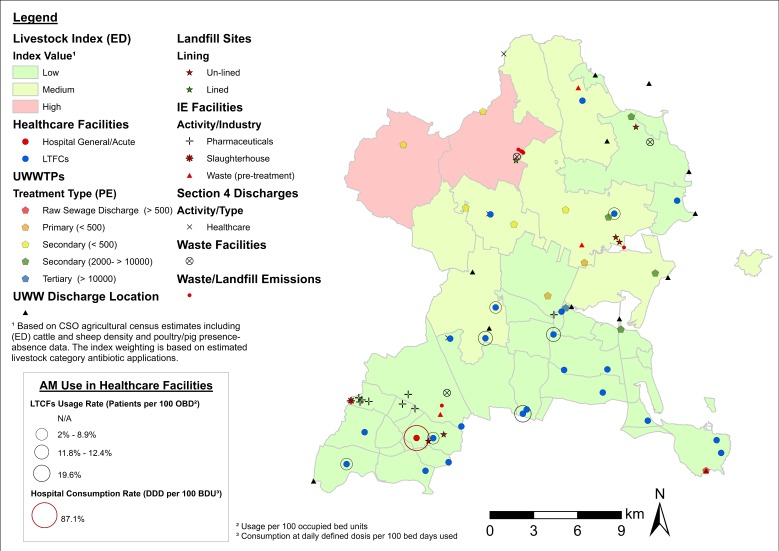
Map of potential ARO sources (classified) and healthcare facilities (including AM use) in Fingal County. Notes: Livestock index estimates are included as a base map layer. ED = electoral district. Blanchardstown—the ward is highlighted. UWWTPs = urban wastewater treatment plants. UWW = urban wastewater. IE = industrial emissions. PE = population equivalent (i.e., estimated population served by UWWTPs).

**Figure 5 antibiotics-08-00016-f005:**
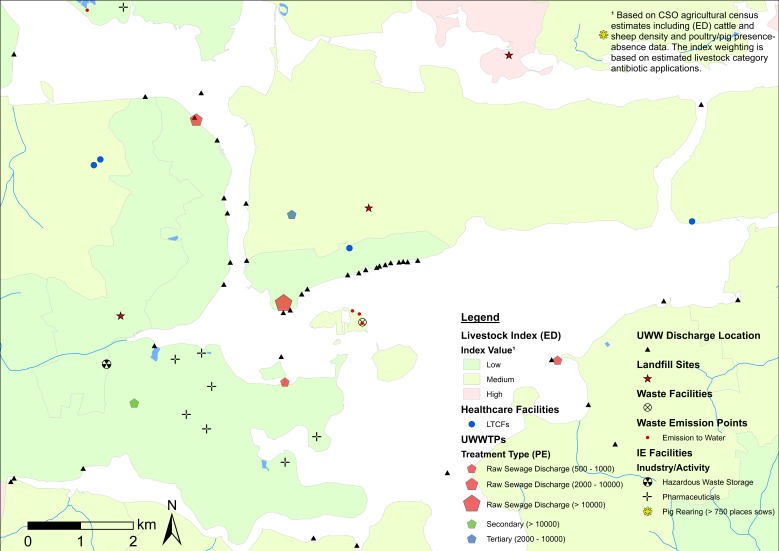
High resolution (fine-scale) map of potential ARO sources (classified) and healthcare facilities in Cork Harbour with livestock index estimates. Notes: Livestock index estimates are included as a base map layer. ED = electoral district. UWWTPs = urban wastewater treatment plants. UWW = urban wastewater. IE = industrial emissions. PE = population equivalent (i.e., estimated population served by UWWTPs).

**Figure 6 antibiotics-08-00016-f006:**
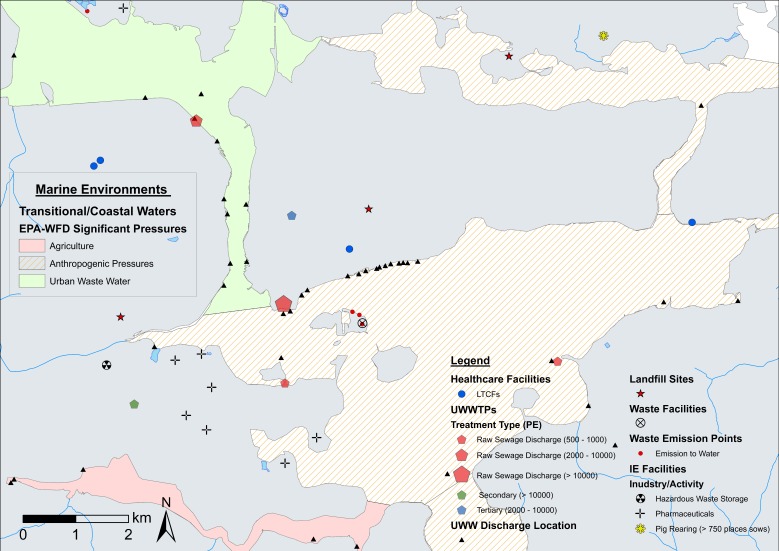
High resolution (fine-scale) map of potential ARO sources (classified) and healthcare facilities in Cork Harbour with coastal/transitional waters significant pressures. Notes: Environmental Protection Agency (EPA)-Water Framework Directive (WFD) coastal/transitional waters significant pressures data layers are also included. UWWTPs = urban wastewater treatment plants. UWW = urban wastewater. IE = industrial emissions. PE = population equivalent (i.e., estimated population served by UWWTPs).

**Figure 7 antibiotics-08-00016-f007:**
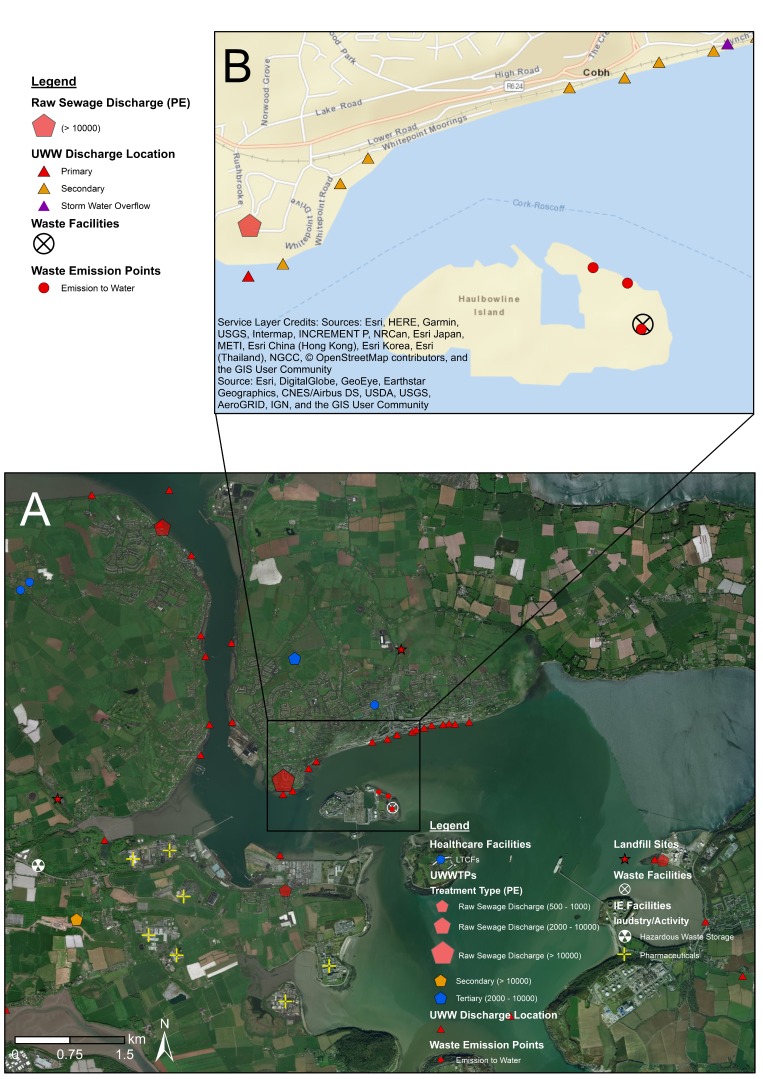
Zoom-in example on a potential sample area using service base map layers (satellite imagery and street map elements) in Cork Harbour. Notes: (**A**) High resolution (fine-scale) map of Cork Harbour including potential ARO sources (classified), healthcare facilities, and satellite imagery base map. (**B**) Zoom-in of Cork Harbour using street map elements base map. UWWTPs = urban wastewater treatment plants. UWW = urban wastewater. IE = industrial emissions. PE = population equivalent (i.e., estimated population served by UWWTPs).

**Figure 8 antibiotics-08-00016-f008:**
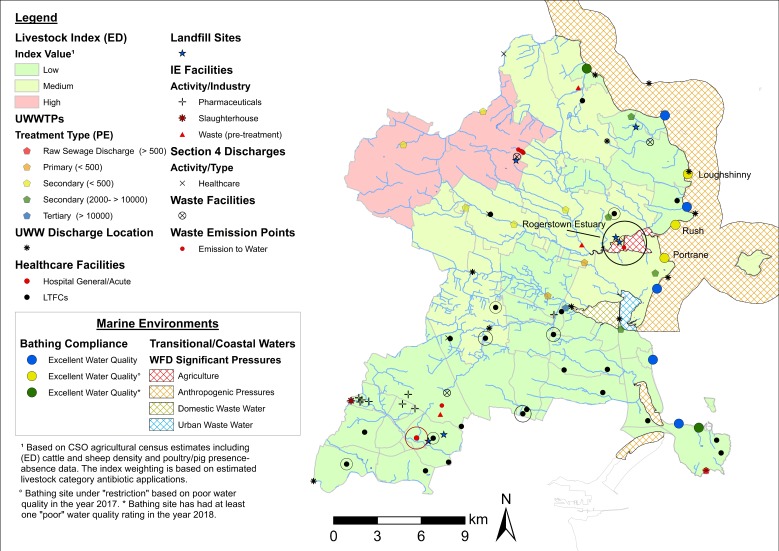
Map of all potential ARO sources and transmission routes in Fingal County. Notes: Livestock index estimates, hydrology (inland waters), EPA-WFD coastal/transitional significant pressures and bathing site compliance data layers are included. Buffers around healthcare facilities reflect rates of AM use/consumption (red = hospital/general acute; black = LTCFs). Relevant areas including the Rogerstown Estuary and bathing sites are highlighted. ED = electoral district. UWWTPs = urban wastewater treatment plants. UWW = urban wastewater. IE = industrial emissions. PE = population equivalent (i.e., estimated population served by UWWTPs).

**Figure 9 antibiotics-08-00016-f009:**
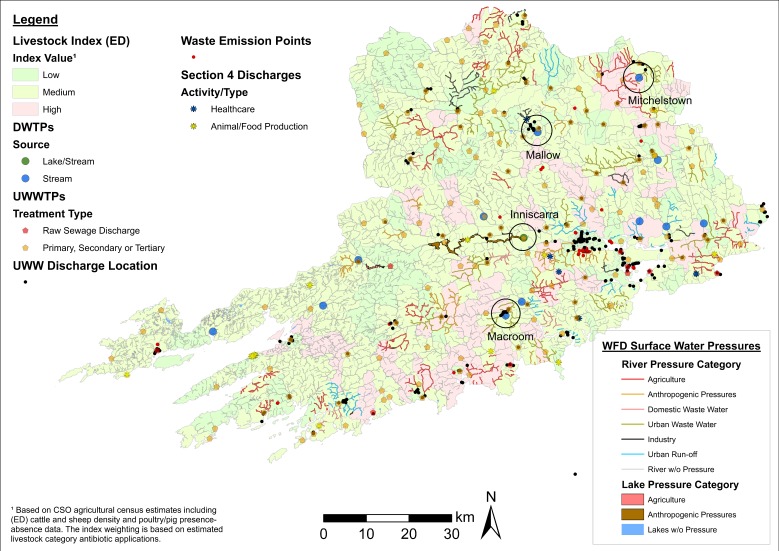
Map of drinking water treatment plants (DWTPs) and selected ARO sources in County Cork. Notes: EPA-WFD rivers/lakes significant pressures and livestock index estimates data layers are included. Relevant DWTPs are delineated with a buffer. ED = electoral district. UWWTPs = urban wastewater treatment plants. UWW = urban wastewater.

**Figure 10 antibiotics-08-00016-f010:**
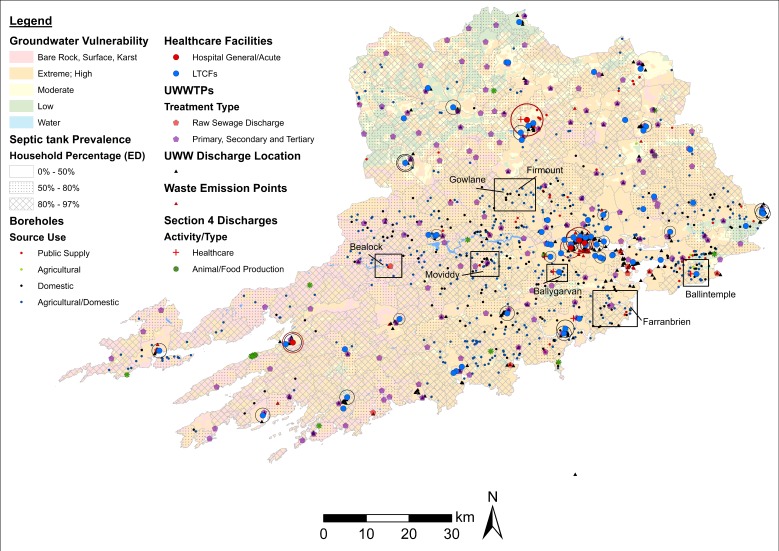
Map of groundwater vulnerability in County Cork. Notes: Boreholes, wastewater AROs sources, and (ED) septic tank prevalence are also included. Buffers around healthcare facilities reflect rates of AM use/consumption (red = hospital/general acute; black = LTCFs). EDs considered suitable for sampling are highlighted. ED = electoral district. UWWTPs = urban wastewater treatment plants. UWW = urban wastewater.

**Table 1 antibiotics-08-00016-t001:** Main themes and key datasets, inclusive of geographic information systems (GIS) shapefiles (feature classes) and attribute data, obtained and collated in the mapping exercise.

Theme	Dataset	Source
Agriculture *	Livestock presence/absence (pig, poultry) *. Livestock numbers (cattle, sheep) *. Livestock density (cattle, sheep) * ^+^. Livestock weighted index * ^+^.	CSO
Demographics *	Population numbers *. Population density * ^+^.	CSO
Groundwater	Groundwater wells (modified to only include domestic/agricultural boreholes). (EPA) Groundwater monitoring stations (including coliform concentrations). **Groundwater vulnerability.** **(EPA-WFD) Significant human pressures on groundwater bodies.** **(EPA-NIP) Groundwater susceptibility to microbial pathogen percolation.** **(EPA-NIP) Groundwater at risk of DWWTS contamination.**	EPA, GSI
Healthcare	General/acute hospitals and long-term care facilities. Antimicrobial usage rate in long-term care facilities °. Antimicrobial consumption in general/acute hospitals °.	HSE, HPSC
Hydrology (or Inland Waters)	**Rivers and lakes.** **(EPA-WFD) Surface water bodies quality and significant human pressures.** **(EPA-NIP) Surface water bodies at risk of DWWTS contamination.**	EPA, OSI
Marine Environments	Aquaculture (finfish farms). **(EPA-WFD) Coastal/transitional water body quality and significant human pressures.** Bathing site compliance (2 year records). Marine dumping sites.	DAFM, EPA
Water Supply	Drinking water treatment plants. Household private well prevalence *.	CSO, EPA
Wastewater	Combined sewage overflows. Integrated constructed wetlands. Raw sewage discharge points. Household septic tank prevalence *. Urban waste water treatment plants (treatment type and PE). Urban waste water discharge locations. Industrial emissions. Integrated pollution control facilities. Section 4 Discharges.	EPA, CSO
Waste Management	Waste and landfill facilities. Waste facility/landfill emissions.	EPA

Notes: Datasets in bold were categorised as primary transmission routes. * attribute data available at the electoral district (ED) level and related to ED (polygon) features. ^+^ generated based on pre-existing attribute data. ° based on point prevalence surveys. EPA = Environmental Protection Agency. WFD = Water Framework Directive. NIP = National Inspection Plan. DWWTS = domestic wastewater treatment systems. CSO = Central Statistics Office. OSI = Ordnance Survey Ireland. GSI = Geological Survey Ireland. HSE = health service executive. HPSC = Health Protection Surveillance Centre. DAFM = Department of Agriculture, Food, and Marine. PE = population equivalent.

**Table 2 antibiotics-08-00016-t002:** Summary statistics for potential sources of antimicrobial resistant organisms (AROs), livestock index estimates, and healthcare facilities identified in the mapping exercise.

LAAs	EDs	ARO Sources	Livestock Index Estimate	Healthcare Facilities
**Galway**	No Specific ED (off-shore)	40	N/A	N/A
	Ballinasloe Urban	16	Low	Hospital (1), LTCFs (2)
	Athenry	11	High	LTCFs (1)
	Gort	9	Low	LTCFs (1)
	Portumna	9	Medium	LTCFs (1)
	Ballynakill	8	Low	N/A
**LAA Total**	-	**350**	**High (10%), Medium (40%),** **Low (50%)**	**Hospitals (3),** **LTCFs (44)**
**Cork**	No Specific ED (off-shore)	138	N/A	N/A
	Lehenagh	33	Medium	N/A
	Killaconenagh	16	Medium	LTCFs (1)
	Tramore (C)	16	Low	N/A
	Mitchelstown	15	Medium	LTCFs (1)
	Bantry Urban	14	Low	Hospital (1), LTCFs (1)
**LAA Total**	-	**930**	**High (16%), Medium (47%),** **Low (37%)**	**Hospitals (5),** **LTCFs (81)**
**Fingal**	No Specific ED (off-shore)	36	N/A	N/A
	Lusk	9	Medium	LTCFs (1)
	The Ward	9	Low	LTCFs (1)
	Blanchardstown-Abbotstown	8	Low	Hospital (1), LTCFs (1)
	Hollywood	7	High	N/A
	Kilsallaghan	7	Medium	LTCFs (2)
**LAA Total**	-	**165**	**High (4%), Medium (16%),** **Low (79%)**	**Hospitals (1),** **LTCFs (24)**

Notes: EDs included in the table represent the top five in terms of number of ARO sources in each LAA. Total estimates at the LAA level are provided in bold and include percentage values of livestock estimates (high, medium, low) calculated from all EDs. Statistics from Galway City are incorporated into Galway County LAA. The no specific ED (off-shore) category refers to ARO sources located in coastal and inland waters. LAAs = local authority areas. ED = electoral district. LTCFs = long-term care facilities. N/A = not available.

**Table 3 antibiotics-08-00016-t003:** Selected examples of thematic maps and data layer combinations generated in the mapping exercise.

Themes	Data Layers	Application
Healthcare, Hydrology, Wastewater.	Healthcare facilities (incl. AM use), rivers/lakes, Section 4 Discharges.	Identify water bodies in close proximity to healthcare facilities and/or in which healthcare facilities discharge effluents.
Healthcare, Hydrology, Water Supply, Wastewater.	Healthcare facilities (incl. AM use), drinking water treatment plants, rivers/lakes, Section 4 Discharges.	Identify drinking water treatment plants in close proximity to healthcare facilities and/or which are fed by water influenced by healthcare facilities effluents.
Healthcare, Groundwater, Water Supply, Wastewater.	Healthcare facilities (incl. AM use), drinking water treatment plants, EPA NIP-WFD groundwater body status *, and wastewater layers (UWW discharges, Section 4 Discharges, UWWTPs).	Identify drinking water treatment plants fed by groundwater at potential risk of contamination from healthcare and wastewater emissions.
Healthcare, Groundwater, Wastewater.	Healthcare facilities (including AM use), boreholes, EPA NIP-WFD groundwater body status *, septic tank prevalence, and wastewater layers (UWW discharges, Section 4 Discharges, UWWTPs, waste emission points).	Identify boreholes extracting groundwater at potential risk of contamination from wastewater, healthcare facilities, and septic tank seepage.
Hydrology, Groundwater, Wastewater.	Rivers/lakes, EPA-NIP surface water bodies at risk of DWWTPs contamination, EPA-NIP groundwater at risk of DWWTPs contamination, industrial emissions, and wastewater layers (UWW discharge location, UWWTPs, waste emission points).	Identify water bodies (surface and groundwater) at potential risk of contamination from DWWTPs, wastewater, and industrial sources.
Groundwater, Wastewater, Water Supply.	Drinking water treatment plants, EPA-NIP groundwater body status *, septic tank prevalence, and wastewater layers (UWW discharge location, UWWTPs, waste emission points).	Identify drinking water treatment plants fed by groundwater at potential risk of contamination from wastewater effluents and septic tank seepage.
Agriculture, Water Supply, Groundwater, Wastewater.	Drinking water treatment plants, EPA NIP-WFD groundwater body status *, livestock index, private well prevalence, and wastewater layers (UWW discharge location, UWWTPs, waste emission points).	Identify drinking water treatment plants and EDs with high private well prevalence (high risk of AROs human exposure) fed by groundwater at potential risk of contamination from wastewater effluents and agricultural emissions.
Agriculture, Hydrology, Water Supply, Wastewater.	ICWs, livestock index, rivers/lakes, and wastewater layers (UWW discharges, UWWTPs).	Identify ICWs located in areas with high livestock estimates and those receiving effluents from UWW and different types of UWWTPs discharges (including treatment type and PE).

Notes: The potential applications in assessing ARO environmental prevalence are also included. * indicates all EPA NIP-WFD groundwater classifications layers (see Table 4) that were used to create sub-sets of thematic maps. AM = antimicrobial. UWW = urban waste water. UWWTPs = urban wastewater treatment plants. DWWTS = domestic wastewater treatment systems. ICWs = integrated constructed wetlands. PE = population equivalent.

**Table 4 antibiotics-08-00016-t004:** Percentage area of groundwater bodies and boreholes under specific EPA NIP-WFD groundwater classifications and categories.

Categories → LAAs ↓	Groundwater Vulnerability					Susceptibility to Microbial Pathogen Percolation				Significant Human Pressures on Groundwater Bodies					
	Rock/Karst	Extreme	High	Moderate	Low	Very High	High	Low	Urban Fabric	Anthropogenic	Agricultural	Domestic Waste Water	Industry	Waste	None
County Galway	15% (15%)	25% (39%)	24% (29%)	14% (12%)	19% (4%)	15% (15%)	26% (38%)	58% (42%)	0.6% (4%)	38% (15%)	19% (53%)	3% (11%)	<0.1% (<0.1%)	<0.1% (N/A)	40% (17%)
County Cork	21% (9%)	33% (30%)	32% (48%)	8% (10%)	6% (3%)	20% (10%)	31% (28%)	47% (56%)	2% (6%)	48% (82%)	3% (1%)	0.2% (2%)	<0.1% (<0.1%)	<0.1% (N/A)	45% (16%)
Fingal County	5% (5%)	5% (5%)	23% (30%)	17% (N/A)	39% (60%)	5% (5%)	13% (N/A)	70% (70%)	12% (25%)	10% (10%)	N/A	N/A	0.3% (N/A)	N/A	89% (90%)

Notes: Boreholes percentage values are provided in parenthesis. Values in bold represent maximum percentage values estimated among all LAAs. Groundwater bodies under ‘water’ categories (i.e., underneath surface water bodies) or with unknown estimates/status are excluded from the table. N/A = not available.

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
