# Peer review of "Mapping and Analysing Potential Sources and Transmission Routes of Antimicrobial Resistant Organisms in the Environment using Geographic Information Systems—An Exploratory Study"

_antibiotics, 2019, doi:10.3390/antibiotics8010016_

Round 1

Reviewer 1 Report

The article details the use of Geographic Information Systems to map potential sources of Antimicrobial Resistant Organisms. The article uses data from from Irish national sources such as the CSO, EPA, and the HSE

Major Comments 

The fundamental issue with the article is that while the methodology is interesting it has not been verified. No sampling has been carried on any of these areas or transmission routes in order to verify if the methodology is correct. No control sampling of areas not deemed to be at risk has been carried in order to verify that the proposed methodology is an accurate indicator.

I would recommend a limited sampling program backed up with statistical analysis in order to determine the accuracy of the methodology.

Some of the selected thematic maps and data layer combinations generated in Table 3 seems to be of little value. For example Healthcare, Hydrology and wastewater. Do many healthcare facilities discharge directly to water sources? Surely most waste from these facilities goes to wastewater treatment plants and is then discharged?  

Minor Comments 

Figure legends in the figures to identify points of interest are small and difficult to read. I would recommend making the figures bigger

Author Response

Point 1: The article details the use of Geographic Information Systems to map potential sources of Antimicrobial Resistant Organisms. The article uses data from from Irish national sources such as the CSO, EPA, and the HSE

Response 1: Thank you for reviewing our paper and for your comments below, which we have now addressed. (in red)

Major Comments

Point 2: The fundamental issue with the article is that while the methodology is interesting it has not been verified. No sampling has been carried on any of these areas or transmission routes in order to verify if the methodology is correct. No control sampling of areas not deemed to be at risk has been carried in order to verify that the proposed methodology is an accurate indicator.

Response 2: Our paper aims to map and analyse potential sources and transmission routes of antimicrobial resistant organisms in the environment. The approach followed consists of a thorough review of the literature that examines relevant sources and transmission routes, as well as the use of geographic information systems (GIS) to map and analyse spatial data on these sources and transmission routes. The aim of this exercise is to illustrate the utility of GIS in gaining insights into the spatial dimension of AMR, as well as to guide sampling campaigns and intervention points. While no sampling is undertaken in this paper, the maps developed in this research are currently being used to inform sampling locations in an Environmental Protection Agency (Ireland) (EPA) funded project. However, results from these sampling campaigns will not be available for a number of years. We are explicit in our paper that the aim of the research in our paper is to help in this regard. Thus, we believe the methodology employed is fully justified. However, to help address the reviewer’s point, we have now re-titled our paper as “Mapping and Analysing Potential Sources and Transmission Routes of Antimicrobial Resistant Organisms in the Environment using Geographic Information Systems”. We have also added the word ‘potential’ on Line 15 in the Abstract and the word ‘likely’ on Line 19 in the Abstract.

Point 3: I would recommend a limited sampling program backed up with statistical analysis in order to determine the accuracy of the methodology.

Response 3: As outlined above, we are currently involved in a large-scale sampling campaign, based on the analysis presented in this paper, the results of which will be not be available for some time. Since one of the main aims of this current paper is to use GIS to guide sampling campaigns, we therefore believe that our methods are appropriate.

Point 4: Some of the selected thematic maps and data layer combinations generated in Table 3 seems to be of little value. For example Healthcare, Hydrology and wastewater. Do many healthcare facilities discharge directly to water sources? Surely most waste from these facilities goes to wastewater treatment plants and is then discharged? 

Response 4: In Ireland it is not the case that all healthcare facilities are necessarily served by wastewater plants. In fact, we do not have data on this (digital sewage network data in Ireland is not available), and thus our mapping exercise can help us to investigate this further by looking at facilities close to raw sewage discharge points, of which there are 38 currently in Ireland. This is particularly the case in more rural areas. Secondly, the mapping of these specific themes will allow us to compare samples of the effluent of wastewater treatment plants and healthcare facilities in the same areas in order to see how efficient the treatment is at removing AROs. By adding the hydrology layer in these maps we can choose recreational seawaters and lakes which may be used as drinking water sources to sample in the vicinity of the healthcare and discharge points to evaluate the risk of environmental and human exposure. In fact, this was precisely the reason for combining these specific layers, which is of primary interest in the AREST project.

Overall, the data combinations presented in Table 3 are meant to familiarize readers with the multiple potential uses of the data collected in this mapping exercise. 

Minor Comments

Point 5: Figure legends in the figures to identify points of interest are small and difficult to read. I would recommend making the figures bigger.

Response 5: The maps will be made available as JPEGs to view online, meaning that interested readers can easily increase the size of the maps on their computer screen, in order to investigate them in greater detail and easily read all of the information which they contain. For maps which contain large amounts of information, as is clearly the case in our paper, this is standard practice.

Reviewer 2 Report

The manuscript describes the use of geographic information systems to track and monitor antimicrobial resistance (AMR), identifying hotspots, distribution, routes of transmission. I think this is an interesing perspective and can be useful to prevent AMR spread and to take more precise measures in the intervention points.

Overall the mansucript is well written and clear. Methods are described and the results well discussed with a good support of references. I enjoy the reading of the manuscript and its given information.

I just have minor comments:

line 45. The majority of antibiotics are not synthetic. They are dereived of natural molecules. It should be replaced by semi-synthetic antibiotics.

Line61. mcr-1 in italic

Author Response

Point 1: The manuscript describes the use of geographic information systems to track and monitor antimicrobial resistance (AMR), identifying hotspots, distribution, routes of transmission. I think this is an interesing perspective and can be useful to prevent AMR spread and to take more precise measures in the intervention points.

Overall the mansucript is well written and clear. Methods are described and the results well discussed with a good support of references. I enjoy the reading of the manuscript and its given information.

Response 1: Thank you for reviewing our paper and for your comments below, which we have now addressed. (in red)

Point 2: I just have minor comments:

line 45. The majority of antibiotics are not synthetic. They are dereived of natural molecules. It should be replaced by semi-synthetic antibiotics.

Response 2a: synthetic changed to semi-synthetic.

Line61. mcr-1 in italic.

Response 2b: mcr-1 changed to mcr-1.

Round 2

Reviewer 1 Report

No suggestions beyond previous comments